# Training for the Future: A Simple Gradient Interpolation Loss to Generalize Along Time

**Anshul Nasery** [*][†]    **Soumyadeep Thakur**[*]    **Vihari Piratla**    **Abir De**   **Sunita Sarawagi**

Department of Computer Science
Indian Institute of Technology, Bombay

## Abstract

In several real world applications, machine learning models are deployed to make predictions on data whose distribution changes gradually along time, leading to a drift between the train and test distributions. Such models are often re-trained on new data periodically, and they hence need to generalize to data not too far into the future. In this context, there is much prior work on enhancing temporal generalization, e.g. continuous transportation of past data, kernel smoothed time-sensitive parameters and more recently, adversarial learning of time-invariant features. However, these methods share several limitations, e.g, poor scalability, training instability, and dependence on unlabeled data from the future. Responding to the above limitations, we propose a simple method that starts with a model with time-sensitive parameters but regularizes its temporal complexity using a Gradient Interpolation (GI) loss. GI allows the decision boundary to change along time and can still prevent overfitting to the limited training time snapshots by allowing task-specific control over changes along time. We compare our method to existing baselines on multiple real-world datasets, which show that GI outperforms more complicated generative and adversarial approaches on the one hand, and simpler gradient regularization methods on the other.

## 1 Introduction

Many organizations operate their machine learning pipeline as follows: collect labeled data based on day-to-day operations e.g. user-clicks for a recommendation model; periodically (say every $\Delta = 48$ hours) train the model on the data collected from the past; deploy the model to serve requests of the next $\Delta$ period; repeat the cycle. Real-world examples of such usage include recommendation models for e-retails, loan approval models in banks, regression models to forecast future popularity of media content, classification of malicious websites, and activity recognition models from sensor data.

A key motivation behind such periodic retraining is the implicit knowledge that the data distribution is not stationary, requiring a revision of the decision boundary with time. While incorporating fresh data in training does make the trained model more current, we investigate if we can improve the training even further. In particular, when training the model at a time $t$, we know that the model will *not* be deployed on data samples in the training interval, but on data from the immediate future $[t, t + \Delta]$ which could differ from even the most recent training data. Our goal in this paper is to use this knowledge more centrally to train a model tailored to the intended deployment period.

**Formal Problem Statement.** We consider prediction tasks over an input space $\mathcal{X}$ and output space $\mathcal{Y}$ where the joint distribution $P(\mathcal{X}, \mathcal{Y})$ evolves with time. The prediction model should therefore be conditioned on time as $P(y|\mathbf{x}, t)$ for $\mathbf{x} \in \mathcal{X}$, $y \in \mathcal{Y}$, and time stamp $t$. During training we are provided with labeled data from $T$ time snapshots $t_1 \leq t_2... \leq t_T$. Call these $\mathcal{D}^1, \mathcal{D}^2, \ldots \mathcal{D}^T$ where each $\mathcal{D}^s = \{(\mathbf{x}_i, y_i) : i = 1 \ldots n_s\}$ is assumed to be sampled from $P(\mathbf{x}, y|t_s)$. The trained model

---

[*]Equal Contribution

[†]anshulnasery@gmail.com

35th Conference on Neural Information Processing Systems (NeurIPS 2021).

will only be deployed on data from a time interval in the future which we denote as $t_{T+1}$, that is on examples sampled from $P(\mathbf{x}|t_{T+1})$. Note that we only evaluate performance once on time $t_{T+1}$, differing from online learning where regret is computed incrementally. Following most practical setups, we assume that the temporal drift across time is not too high. Since the deployment time is in the future, we cannot assume the presence of even unlabeled target data at the time of training the model. Further, at different time snapshots, completely unrelated sets of $\mathbf{x}_i$ may comprise a $\mathcal{D}^s$. This makes the problem much harder than standard time-series forecasting tasks, since we do not have values of the same instance at different points in time. Our goal is not to generalize to all future data, but rather to data relatively close in the future, since our motivating application is one where models are retrained periodically. The former generalization would require very different models and approaches than what we propose. We assume that $P(y|\mathbf{x}, t)$ is implemented as a general-purpose neural network $F(\mathbf{x}, t)$ that takes as input $\mathbf{x}, t$ and outputs either (1) a vector of logits for classification tasks or (2) mean and other scores defining a continuous distribution over predicted $y$ for regression tasks.

A solution often used in practice is to only train the model on recent data and/or apply a graceful ageing penalty on past data. However, this may not be adequate with data hungry modern networks, especially when non-obvious seasonality makes distant data relevant. So we seek general purpose solutions that can harness all available data for making optimal decisions into the future. Any such solution first needs to make the model time sensitive. This can be done by feeding time as an input feature, or by making all or some of the model parameters to be a function of time. However, that by itself cannot be sufficient to generalize to future data if we train the model to minimize loss on the training snapshots. In fact, a sufficiently complex model can easily overfit on training time-stamps when the number of instances per snapshot is large and the number of time stamps $T$ is small.

The field of continuous learning [1] offers one approach to tackle this problem. However, in our case the training data can be revisited and fresh models retrained making issues like catastrophic forgetting irrelevant. More relevant to our problem formulation are recent continuous domain adaptation methods [2, 3, 4, 5, 6] which either try to *predict* how the training domains evolve as time passes, or learn feature representations which are invariant across time [4, 5]. However, these domain adaptation approaches assume the presence of unlabeled data in the future, which is not available in our case. Another set of approaches aim to make network parameters a function of time [7, 8, 9] but they require external smoothness kernels as hyper-parameters for each parameter-type, which cannot be easily trained end-to-end on deep neural networks.

The key objective of our desired training algorithm is to generalize to the near future, even at the cost of performance on past data. In order to achieve this goal, the training algorithm needs to encourage the model to adjust for temporal drifts both in the data distribution and the decision boundary, and be able to generalize and extrapolate well taking into account such shifts. To this end, we aim to develop a supervised learning model which would be robust to small perturbations in time— which is fed as feature. We attempt learning functions that are smooth in time, ensuring a limited temporal drift from past to future. However, given a limited number of training time-stamps, a moderately complex model is prone to fit the training data perfectly, without generalizing it well to the future. To tackle this problem, we propose a method which can learn to generalize well in time by interpolating the predictor between two training time steps using a first order Taylor series expansion. This interpolated predictor is supervised using the available labels. While doing so, it makes optimal use of the temporal information, which allows it to extrapolate well on future time steps.

Our gradient interpolation method gives us the privilege to use a complex temporal model even if the number of timestamps is small without the fear of overfitting. To that end, we develop a novel neural model that captures the evolving decision boundary over time. More specifically, we design a novel time dependent activation function called leaky Temporal ReLU (TReLU), which changes the underlying activation threshold smoothly along time. Such a model along with the Gradient Interpolation (GI) based loss function allows our method to make the optimal use of the temporal information and thus extrapolate well to future time.

**Our Contributions.** In this work, we propose a simple training algorithm to encourage a model to learn functions which can extrapolate well to the near future. We do this by supervising the first order Taylor expansion of the learnt function. We also propose a method to make neural networks time aware by modifying the leaky ReLU activation function to change with different time-stamps. We demonstrate strong empirical performance of our method by comparing against state-of-the-art methods and practical baselines on several real world datasets, and also give insights into its workings through a simple synthetic setup.

## 2 Related Work

As we discussed in Section 1, our work is closely related to continuous domain adaptation where time can be treated as a continuous valued domain index. Approaches tackling such problems can be broadly classified into three categories: (1) biasing the training loss towards future data via transportation of past data, (2) using time-sensitive network parameters and explicitly control their evolution along time, (3) learning representations that are time-invariant using adversarial methods. The first category augments the training data, the second category reparameterizes the model, and the third category redesigns the training objective. We discuss each of these in detail.

**Transportation Methods** Transformation based approaches [10, 11] map data samples from the past distribution into the target distribution, and train a classifier on the transported samples. These have been adapted to situations in which the domains change continuously in [3] and [2]. [3] proposes *CDOT*, which applies optimal transport with a regularizer to take into account the similarity between two successive distributions. [2] modifies the approach to work in an environment where the target data is given as a stream, which is continually changing. To compute the transformations, these methods require a similarity measure on $\mathbf{x}$, along with unlabelled data from the target domain, which makes them unsuitable for our settings. Further, the transformations induced may be too simple for high-dimensional inputs $\mathbf{x}$, and may not scale on large training sets.

**Parameter smoothness** Another approach is to make the model parameters a function of time, and to control their smoothness along time. Mancini et. al. in [9] propose *Adagraph*, a deep-neural network for image data where the scale and shift batch norm parameters are domain-specific. They propose using a user-specified kernel to capture the relationship between the domains, and computing the test domain-specific parameters using a kernel regression on the train domain parameters. This kernel is a hyper-parameter and needs to be carefully designed. The method cannot hence discover relationships between domains in an end-to-end fashion. Other methods like [7] and [8] consider only a logistic regression classifier and model the parameters as stochastic processes like vector autoregression (VAR) or a Gaussian process (GP). These methods require computing the posterior of GP-smoothed parameters, and this inference does not scale for multi-layered neural networks.

**Time invariant representations** A recent state of the art approach for continuous domain adaptation is *CIDA* [5] that takes time as input but requires the neural network to encode $[\mathbf{x}, t]$ as a time-invariant feature vector to be passed to a common classifier. The encoder is trained using standard two-party adversarial game where the discriminator tries to recover time using a regression loss, and the encoder attempts to suppress time. It is well-known that training such two-party games is challenging and does not converge easily. Even when such convergence is achieved, we argue that a better strategy is to allow the features to evolve with time, while controlling the smoothness of this change.

**Domain Generalization** The goal of such methods is to generalize to an open unordered set of domains given multi-domain data during training but no unlabeled data during deployment. Several methods have been proposed for this task including domain invariant networks [12], decomposition methods [13], meta-learning [14] or data augmentation with domain perturbations [15]. Our problem can be considered a special case of domain generalization where the domains are ordered, and we only care about generalizing to an adjacent closed interval.

Another emerging area deals with settings where the labeled data is from a fixed domain but the target domain evolves with time and is unlabeled [16, 17]. This is different from our setup where we assume evolving labeled data in the source, and one target domain without any data.

**Online Learning** [18] Also called continual learning [19] and life-long learning, attempts to learn models using training data that arrives in a stream. These have been extended to deep networks [20] using techniques like meta-learning [21] and randomized stochastic gradient descent [22]. Methods such as FTPL and FTRL [23] could also be extended to provide an alternate approach to the problem. However, online methods do not allow revisiting the training data, unlike in our case.

In addition to the above techniques we explored a number of other strategies for augmenting training data in the future in addition to transportation methods. We devised a generative model and a local perturbation method inspired by [24, 15]. We also explored the use of neural ODEs for transporting data along time since these are specifically meant to capture continuous dynamics. These approaches are discussed in detail in secction A.2 of the supplementary. None of these approaches provided as much consistent gains on a variety of data types on modern neural networks, as this surprisingly simple gradient interpolation method that we describe in the next section.

# 3 The Gradient Interpolation Approach

Our approach comprises of two parts: (1) modelling $F(\mathbf{x}, t)$ as a time sensitive neural network $F_\theta(\mathbf{x}, t)$ with $\theta$ being the trainable parameters and (2) imposing a special loss to encourage the network to generalize to the near future. We make the network time-sensitive by both taking time as an additional input concatenated with $\mathbf{x}$, and by computing a subset of the network parameters as a function of time. When a parameter $W_j$ of a network is made time-sensitive, then $W_j$ becomes the output of a small neural network $w_j(t|\theta_j)$ with time as input and parameters $\theta_j$. Now instead of $W_j$ we train $\theta_j$ during the end-to-end training. The set of $W_j$s that are made time-sensitive is part of network architecture design. We present our proposed design of a time-sensitive network architecture in Section 3.3. The rest of our algorithm treats the $\theta_j$s as any other network parameter and does not impose any custom smoothing on them unlike some prior work [8, 9].

## 3.1 Training loss induced by gradient interpolation

**Overfitting issues for time-sensitive network.** Despite building a time-sensitive network, if we apply standard ERM training on data from $T$ training snapshots $D^1, \ldots, D^T$, we have no guarantee that the network would generalize to samples from the $P(\mathbf{x}|T+1)$ distribution, since the network parameters could easily overfit individually on the $T$ training distributions. Responding to this challenge, we propose to regularize the output of the entire network, as described below

**Introducing gradient interpolation induced training loss.** We encourage learning functions that can be approximated linearly by regularizing the curvature of $F_\theta(\mathbf{x}, t)$ along time. We impose a loss on the first order Taylor approximation of the model output. Formally, if $\ell(\cdot; \cdot)$ denotes a loss metric to be minimized, our gradient interpolated loss is defined as

$$\mathcal{J}_{GI}(y; F_\theta(\mathbf{x}, t)) = \ell(y; F_\theta(\mathbf{x}, t)) + \lambda \max_{\delta \in (-\Delta, \Delta)} \ell\left(y; F_\theta(\mathbf{x}, t - \delta) + \delta \frac{\partial F_\theta(\mathbf{x}, t - \delta)}{\partial t}\right) \quad (1)$$

Here, the first term is the usual loss for fitting $F_\theta(\mathbf{x}, t)$ to $y$, while the second term is the loss on a regularized approximation of $F_\theta(\mathbf{x}, t)$ using the Taylor Expansion at $t - \delta$. We choose the $\delta$ adversarially by gradient ascent within a user-provided window $\Delta$, which we set as a task-dependent hyper-parameter. When $\delta$ is negative we are computing the logits of an instance at $t$ by extrapolating from the future $t + \delta$. This indirectly serves to supervise the function and its gradients to near-by points. We thus expect this loss to provide better generalization to near future points in the neighbourhood of the last training time step.

We summarize our training procedure in Algorithm 1. It works in three steps. Given a parameterized neural model $F_\theta$ it first pretrains $\theta$ using ERM loss on the entire dataset (for loop in lines 3–4). Then, for each minibatch in each time snapshot, we first adversarially select a $\delta$ by gradient ascent to maximize the loss $\ell(\cdot; \cdot)$ and update the model parameters $\theta$ defined in Eq. (1) using $\delta$ (for loop in lines 9–15). Moreover, while doing so, we always clamp the value of $\delta$ in $(-\Delta, \Delta)$, as suggested in Eq. (1).

---

**Algorithm 1** GI based training algorithm to estimate $F$

---

**Require:** Dataset $\{\mathcal{D}^s = \{\mathbf{x}_i, y_i\}, s \in [T]\}$, time stamps $\{t_s \in [T]\}$, a neural model for $F$, i.e., $F_\theta$, initial model parameters $\theta_0$, learning rate $\gamma$, and number of finetune domains $k$.

1: /* Pre-train $F_\theta(\bullet, \bullet)$ on each $\mathcal{D}^s$ */
2: $\theta \leftarrow \theta_0$
3: **for** $s \in [T]$ **do**
4:     $\theta \leftarrow$ PRETRAIN$(\theta, \mathcal{D}^s)$
5: **end for**

6: /* Train $F_\theta(\bullet, \bullet)$ using loss (1) */
7: **for** $s \in [T - k, T]$ **do**
8:     $\{\mathcal{B}\} \leftarrow$ SPLITINBATCHES$(\mathcal{D}^s)$
9:     **for** each minibatch $\mathcal{B}$ **do**
10:         $\delta_0 \leftarrow$ Unif.$[-\Delta, \Delta]$
11:         $\delta \leftarrow$ TRAINDELTA$(\delta_0, \mathcal{B})$
12:         $\delta \leftarrow$ CLAMP$(-\Delta, \Delta)$
13:         $\mathcal{J}_{GI} \leftarrow$ COMPUTELOSS$(\theta, \delta, \mathcal{B})$
14:         $\theta \leftarrow \theta - \gamma \nabla_\theta \mathcal{J}_{GI}$
15:     **end for**
16: **end for**
17: **return** $\theta$.

---

## 3.2 Understanding GI with a synthetic setup

We present here an empirical analysis with a simple setup to understand how GI could be more effective in generalizing to the future compared to existing methods. Consider a dataset where the input is $d$ Boolean features i.e., $\mathcal{X} \in [0, 1]^d$, and output label $\mathcal{Y} \in [0, 1]$ is binary. Assume their joint distribution factorizes as $\Pr(\mathbf{x}, y|t) = \Pr(y) \prod_{j=1}^d \Pr(x_j|y, t)$ where prior class probability is

independent of time and $\Pr(y) = \Pr(y|t) = 1/2$, and features are conditionally independent given the class label. Assume $\Pr(x_j|y,t) = \Pr(x_j = y|t) = p_j(t)$ where $p_j(t)$ denotes the time-varying correlation between feature $x_j$ and label $y$ at time $t$. Further, let $p_j(t)$ be defined as (for $j \geq 3$):

$$p_1(t) = 0.6 + 0.1t, \quad p_2(t) = 0.6, \quad p_j(t) = 0.5 + 0.49[[t = j - 3]]$$

Here the first feature's correlation with class label evolves linearly with time, the second feature has a constant but slightly positive correlation, whereas each of the remaining features have a strong correlation at exactly one time stamp, and are uninformative elsewhere. Given the conditional independence of features, the following parametric form can perfectly fit the data while allowing for easy interpretation:

$$F_\theta(\mathbf{x}, t) = \sum_{j=1}^{d} w_j(t|\theta_j)x_j + w_0(t|\theta_0)$$

where the $w_j(t|\theta_j)$ can be any neural network with input $t$ and parameters $\theta_j$. We used a two layer network with ReLU activation of width 50 and 20 respectively for each $w_j(t|\theta_j)$.

Assume we have training data at $t = 0, 1, 2$ and we need to deploy at $t = 3$. According to the above parameterization, the baseline ERM could fit training data well by giving a high weight to $x_3, x_4, x_5$ at times $t = 0, 1, 2$ respectively, and low weight at other times. There is no control on how the $w_j(t)$ values evolves with time, and there is no reason for the classifier to generalize to $t = 3$. In fact here the test accuracy is 49.5% (slightly worse than random) while the train accuracy is 99%. Figure 1 shows the weights as a function of time (solid lines) for $w_1, w_2, w_4, w_5$ and we see that $w_2$ is largely ignored while $w_4, w_5$ take large values and have large swings.

When GI is applied on this dataset, we impose an additional loss on logits computed from $\sum_j x_j[w_j(t - \delta) + \delta \frac{\partial w_j(t-\delta)}{\partial t}]$. This loss provides supervision on not just the parameter values at the training times, but also requires that their gradients be correctly oriented to allow reconstruction at near-by points. This encourages information sharing between training time-stamps for the model, and has the effect of smoothing the $w_j(t)$ functions for the transient features, and recovering the correct slope of the linear features. Further, when $\delta$ is negative then for instances close to $T$ we also indirectly supervise the logits and their gradient at future times $T + \delta$.

Figure 1 shows the weights learned by GI in contrast with those of ERM. The accuracy obtained by GI is 70.9, which is higher than that of any other model. Without clairvoyance, and without access to even unlabeled data from $t = 3$, it may be difficult to get better results in expectation over future datasets.

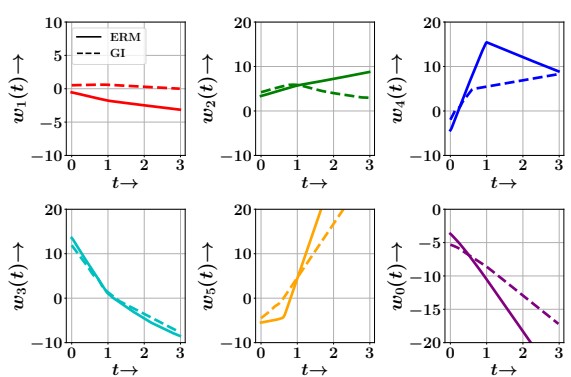

Figure 1: Weights learnt as a function of time

**Is GI loss just a gradient regularizer?** Since one of the main outcomes of GI is smoothing $w_j(t)$s, a natural question is whether the benefits of GI could also be obtained by a simple regularizer on the gradient of the network with respect to time. In that context we explored an alternative gradient penalized training objective as follows:

$$\mathcal{J}_{GR}(y; F_\theta(\mathbf{x}, t)) = \ell(y; F_\theta(\mathbf{x}, t)) + \lambda \left\| \frac{\partial F_\theta(\mathbf{x}, t)}{\partial t} \right\|_2^2 \tag{2}$$

This objective requires us to carefully select the weight $\lambda$ of the regularizer. If $\lambda$ is large, the objective will converge to constant $w_j(t)$, even for weights that have a predictable linear change along time. Such a classifier gives a test accuracy of $47.4\%$ in this setting. For small $\lambda$, the weights would not be sufficiently smooth. Contrast the above with the GI loss in Equation 1 where the true label $y$ supervises how much the gradient needs to be smoothed along time.

**Why time-invariant representation would not work.** GI allows the last layer features to evolve with time, unlike recent methods such as CIDA [5] that attempt to learn time-invariant representations at the last layer. We implement this method by making the vector $\boldsymbol{w}(t) \odot \mathbf{x}$ (where $\odot$ denotes element-

wise product) as the time invariant representation of a data point. A successful implementation of the adversarial loss would make the expected value of $\mathbb{E}_x[w_j(t)x_j]$ independent of time. In our generative model, it can be see that the marginals $\mathbb{E}_x[x_j] = 1/2$, and thus are independent of time. This implies that an ideal classifier as per CIDA would make $w_j(t)$ a constant independent of time. When such a classifier is fit on the data, we get a test accuracy of 50.5% which is again random. The merit of GI is that it can let the feature representation evolve with time to generalize to the future interval.

### 3.3 Our Proposed Time-sensitive Model Architecture

In this section, we present our proposed time-sensitive model architecture $F_\theta$ which approximates $F$ using a neural network model parameterized with $\theta$. Given the feature vector $\mathbf{x} \in \mathcal{D}$ at a time snapshot $t$, the predictor $F_\theta(\mathbf{x})$ uses two time-sensitive components:

**Time representation vector.** First, we featurize the time $t$ associated with the data point $\mathbf{x}$. A naive way to do that is to concatenate $t$ with $\mathbf{x}$ to obtain an augmented feature vector $[\mathbf{x}, t]$. However, such an approach cannot capture complex trends in data, e.g., periodicity. To tackle this challenge, we obtain a representation vector $\boldsymbol{\tau}_t \in \mathbb{R}^m$ of time $t$ by building upon an existing time embedding model TIME2VEC proposed in [25], which is computed as,

$$\boldsymbol{\tau}_t[a] = \begin{cases} \omega_a\, t + b_a, & 1 \le a \le m_p \\ \sin(\omega_a\, t + b_a), & m_p < a \le m \end{cases} \tag{3}$$

Hence the first $m_p$ entries of $\boldsymbol{\tau}$ use a linear transformation of $t$, whereas the last $m - m_p$ entries use a periodic transformation. Here, $\{\omega_a\}$ and $\{b_a\}$ are trainable parameters.

**TReLU activation function.** Another way to make $F$ time-evolving is to model the feature weights in a time dependent manner. Previous work model such weights using Gaussian Processes [8], batch normalization using kernel regression [9], etc. However, extensions of these models in deep neural network may be computationally infeasible for training.

We note that many successful universal approximators often consists of a cascade of linear and ReLU layers. Therefore, in order to encapsulate the time-evolving nature of $F$, we ideally require a universal approximator of time-dependent function. To this aim, we develop a novel time dependent ReLU function, called as TReLU, as follows.

$$\text{TReLU}_\phi(\mathbf{x}, t) = \begin{cases} \boldsymbol{h}_\phi(\boldsymbol{\tau}_t) \odot (\mathbf{x} - \boldsymbol{g}_\phi(\boldsymbol{\tau}_t)) + \boldsymbol{v}(\boldsymbol{\tau}_t), & \text{when } \mathbf{x} < \boldsymbol{g}_\phi(\boldsymbol{\tau}_t) \\ \mathbf{x}, & \text{when } \mathbf{x} \ge \boldsymbol{g}_\phi(\boldsymbol{\tau}_t) \end{cases} \tag{4}$$

Note that, $\boldsymbol{h}_\phi \in \mathbb{R}^d$, $\boldsymbol{v}_\phi$ and $\boldsymbol{g}_\phi \in \mathbb{R}^d$ are neural networks with parameters $\phi$, which control the slope and threshold. If $\boldsymbol{h}_\phi = \mathbf{1}$ and $\boldsymbol{v}_\phi = \boldsymbol{g}_\phi = \mathbf{0}$, TReLU becomes equivalent to traditional ReLU activation function. We then use these TReLU units as a drop in replacement for all the ReLU units in a neural network. We compute the output $y$ for the input $\mathbf{x}$ by successively passing it through linear and TReLU layers according to the network architecture. Note that each TReLU unit in the network is parameterized by a separate set of parameters $\phi$.

Note that the outputs of $\boldsymbol{h}_\phi$ and $\boldsymbol{g}_\phi$ are $d$ dimensional. In our experiments, we use neural networks with single hidden layer of size $k = 8$ units to parameterize these functions, and also consider the dimension of time vectorization, i.e. $m$ to be 8. This introduces $2(mk + kd) = 128 + 16d$ extra parameters per TRelU layer into the network, which is much less than $\mathcal{O}(d^2)$ parameters of the linear layer in the network. This is also less than $\mathcal{O}(Td)$ extra parameters introduced by Adagraph [9] to model temporal dependencies.

Finally $\theta = \{\omega_\bullet, b_\bullet, \phi_\bullet\}$, are estimated using the training algorithm shown in Algorithm 1.

## 4 Experiments

We compare GI with several existing approaches on seven datasets and present a detailed ablation study. All results are reported across five random restarts. Our experimental setup is coherent with our problem setup as described in Section 2. The datasets under consideration have a temporal component associated with them. We consider data with timestamps $1, 2, \ldots, T - 1$ to be our train data, and data from $T$ to be the test data. We train all models on the training data, and report the

| | Classification | | | | | | Regression | | |
|---|---|---|---|---|---|---|---|---|---|
| Method | 2-Moons | Rot-MNIST | ONP | Shuttle | Reuters | Elec2 | House | M5-Hob | M5-House |
| Baseline | $22.4 \pm 4.6$ | $18.6 \pm 4.0$ | $\mathbf{33.8 \pm 0.6}$ | $0.77 \pm 0.10$ | $8.9 \pm 1.02$ | $23.0 \pm 3.1$ | $11.0 \pm 0.36$ | $0.27 \pm 0.10$ | $0.24 \pm 0.08$ |
| LastDomain | $14.9 \pm 0.9$ | $17.2 \pm 3.1$ | $36.0 \pm 0.2$ | $0.91 \pm 0.18$ | $8.6 \pm 1.10$ | $25.8 \pm 0.6$ | $10.3 \pm 0.16$ | $2.72 \pm 0.75$ | $3.17 \pm 1.54$ |
| IncFinetune | $16.7 \pm 3.4$ | $10.1 \pm 0.8$ | $34.0 \pm 0.3$ | $0.83 \pm 0.07$ | $8.7 \pm 0.81$ | $27.3 \pm 4.2$ | $9.7 \pm 0.01$ | $0.12 \pm 0.05$ | $0.17 \pm 0.10$ |
| CDOT | $9.3 \pm 1.0$ | $14.2 \pm 1.0$ | $34.1 \pm 0.0$ | $0.94 \pm 0.17$ | $10.5 \pm 1.10$ | $17.8 \pm 0.6$ | - | - | - |
| CIDA | $10.8 \pm 1.6$ | $9.3 \pm 0.7$ | $34.7 \pm 0.6$ | DNC | DNC | $\mathbf{14.1 \pm 0.2}$ | $9.7 \pm 0.06$ | $0.40 \pm 0.07$ | $0.58 \pm 0.11$ |
| Adagraph | $8.0 \pm 1.1$ | $9.9 \pm 1.0$ | $40.9 \pm 0.6$ | $0.47 \pm 0.04$ | $7.9 \pm 0.51$ | $20.1 \pm 2.2$ | $9.7 \pm 0.10$ | $1.64 \pm 0.28$ | $0.87 \pm 0.14$ |
| GI | $\mathbf{3.5 \pm 1.4}$ | $\mathbf{7.7 \pm 1.3}$ | $34.9 \pm 0.4$ | $\mathbf{0.29 \pm 0.05}$ | $\mathbf{7.4 \pm 0.28}$ | $16.9 \pm 0.7$ | $\mathbf{9.6 \pm 0.02}$ | $\mathbf{0.09 \pm 0.03}$ | $\mathbf{0.05 \pm 0.02}$ |

Table 1: Comparison of our proposed method against existing methods on all the nine datasets in terms of misclasssication error (in %)for first six datasets and mean absolute error (MAE) for last three datasets. The standard deviation over five runs follows the $\pm$ mark. DNC indicates that the method did not converge. We observe that GI outperforms almost all the baselines.

performance on the test data. For tuning hyperparameters, we consider data from $T - 1$ to be the validation set. Code for reproducing our experiments has been open-sourced[3].

## 4.1 Datasets

We experiment with six classification datasets: Rotated 2 Moons (2-Moons), Rotated MNIST (Rot-MNIST), Online News Popularity (ONP) [26], and Electrical Demand (Elec2), Shuttle and Reuters; and three regression datasets: House prices dataset (House), M5-Hobbies (M5-Hob) and M5-Household (M5-House). The first two datasets are synthetic where the angle of rotation is used as a proxy for time. The remaining are real world temporally evolving datasets. Details of the datasets are available in section A.1 of the Appendix.

## 4.2 Methods compared

We compare our method with three practical baselines and three state of the art continuous domain adaptation methods.

**Baseline.** This time-oblivious model is trained using ERM on all the source domains.

**LastDomain.** This time-oblivious model is trained using ERM on the last domain.

**IncFinetune.** In this model, we bias the training towards more recent data by applying the Baseline method described above on the first time-stamp and then, fine-tuning with a reduced learning rate on the subsequent time-stamps in sequential manner. However, the underlying network is otherwise not time-sensitive. Such an approach is similar in spirit to the gradual domain adaptation method of [16], modified to account for labelled data at each time snapshot. This is also similar to online learning methods adapted for our setting.

**CDOT [3].** transports most recent labeled examples $\mathcal{D}^T$ to the future using a learned coupling from past data, and trains a classifier on them.

**CIDA [5].** This method is representative of typical domain erasure methods applied to continuous domain adaptation problems.

**Adagraph [9].** This method makes the batch norm parameters time-sensitive and smooths them using a given kernel.

More details about these methods can be found in section A.2 of the appendix. Comparisons with modified online learning methods from [23] are also present in section C of the appendix.

## 4.3 Comparative analysis

We first compare the performance of our proposed GI based method against existing approaches, in terms of misclasssication error for first six datasets, i.e., 2-Moons, Rot-MNIST, ONP, Elec2 and mean absolute error (MAE) for last three datasets, i.e., House, M5-Hob and M5-House. Table 1 summarizes the results. In most cases, we see that GI provides significantly lower errors. The gains we obtain on M5-House and M5-Hob are particularly impactful because these are large real-life datasets and the gains over the second-best method are statistically significant by a big margin. Additionally, we make a number of other interesting observations from these detailed experiments: (i) All datasets except

---

[3]https://github.com/anshuln/Training-for-the-Future

ONP exhibit strong temporal drift as seen by the poor performance of the time-oblivious Baseline, (ii) More data helps and heuristically choosing a suffix of available data may be sub-optimal as seen by the mixed results on the simple LastDomain baseline. (iii) Just incremental fine-tuning where most recent data is seen last by the model is often a strong baseline. (iv) The CIDA method that creates time-invariant representations shows improvements but in two cases it is worse than IncFinetune.

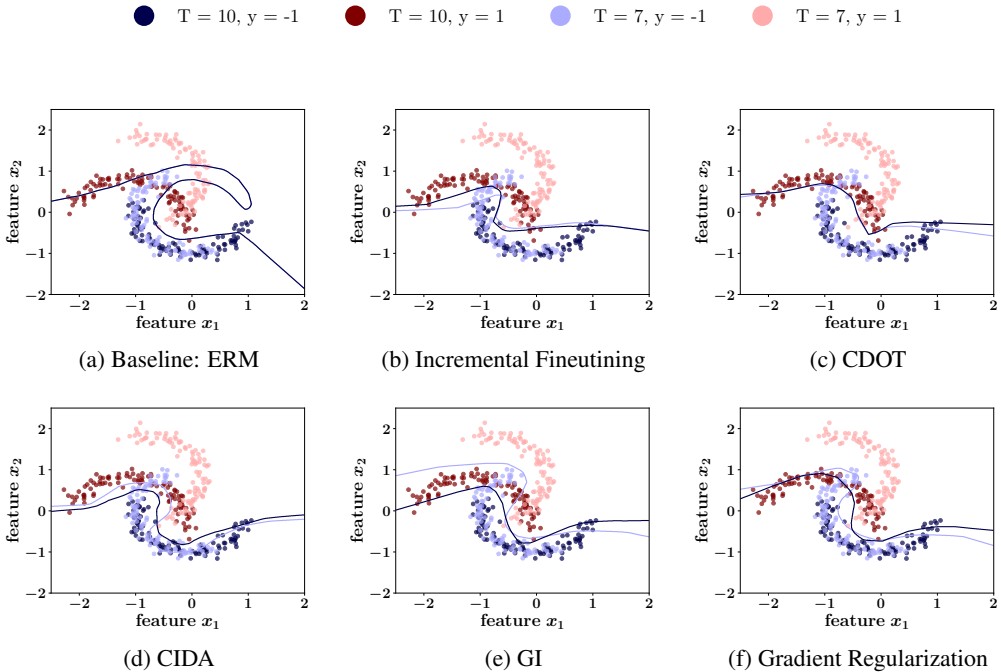

Figure 2: Qualitative Analysis on 2-moons across different methods. Pink (+1) and light purple (-1) are samples from $T = 7$, and magenta(+1) and dark blue (-1) are samples from $T = 10$ differing by a $54°$ rotation. Decision boundaries at $T = 10$ are in dark blue lines and $T = 7$ are in light blue lines.

## 4.4 Qualitative Analysis on 2-moons

Next we compare different methods in a qualitative manner. Figure 2 displays the results, which shows the decision boundary learned by the different classifiers at $T = 10$ and $T = 7$ time-stamps on the 2-Moons dataset. We observe that the time-agnostic baseline overfits the training data and shows poor performance for $T = 10$. Furthermore, IncFinetune significantly improves on the baseline but seems to be over-fitted for the last timestamp. We also note that performance of CDOT depends on how closely the transported images resemble the true target image. Since the transported samples are quite noisy, it does not perform quite as well as other methods that model the dependence on time directly. CIDA is better but misclassifies the outer ends. Gradient regularization makes the decision boundary almost time-invariant, significantly hurting performance due to lack of generalizability to the future. On the other hand, GI is the only method that manages to learn a decision boundary of the correct complexity, and rotates correctly along time.

## 4.5 Ablation study

GI differs from existing methods in two ways: (1) time-sensitive network with time as input via a Time2Vec encoding and TReLU, and (2) the gradient interpolated loss. Here, we perform an ablation study to investigate which of these components contribute the most to GI's gains and also discuss if simpler regularizers could achieve what our GI loss does. More specifically, we consider four alternatives to compare to our proposed method. (1) Base-Time: Here we make the network architecture time-sensitive in the same way as in GI. However, we train using simple ERM. (2) IncFinetune-Time: Here, we bias the time-sensitive network towards recent data using incremental fine-tuning. (3) Grad-Reg: Here we regularize the L2-norm of the gradient of the model along time as in Equation 2, (4) TimePerturb: Here we adapt the popular adversarial training method to make a

| Ablation | 2-Moons | Rot-MNIST | Elec2 | Shuttle | M5-Hob | M5-House |
|---|---|---|---|---|---|---|
| Base-Time | $4.1 \pm 0.87$ | $10.3 \pm 0.9$ | $18.5 \pm 1.7$ | $0.61 \pm 0.14$ | $0.35 \pm 0.09$ | $0.29 \pm 0.14$ |
| IncFinetune-Time | $6.9 \pm 3.3$ | $9.2 \pm 0.9$ | $19.9 \pm 1.4$ | $0.52 \pm 0.12$ | $0.10 \pm 0.04$ | $0.07 \pm 0.02$ |
| Grad-Reg | $11.2 \pm 3.46$ | $11.5 \pm 1.5$ | $26.3 \pm 1.8$ | $0.73 \pm 0.15$ | $0.90 \pm 0.56$ | $2.57 \pm 1.01$ |
| TimePerturb [24] | $\mathbf{3.3 \pm 0.40}$ | $9.9 \pm 0.7$ | $17.3 \pm 0.6$ | $0.67 \pm 0.06$ | $\mathbf{0.09 \pm 0.01}$ | $0.11 \pm 0.04$ |
| GI | $3.5 \pm 1.37$ | $\mathbf{7.7 \pm 1.3}$ | $\mathbf{16.9 \pm 0.7}$ | $\mathbf{0.29 \pm 0.05}$ | $\mathbf{0.09 \pm 0.03}$ | $\mathbf{0.05 \pm 0.01}$ |

Table 2: Ablation study. Comparison of performance between our method and four alternatives across four datasets for classification tasks and two datasets for regression tasks.

model robust to small changes in time [24]. In particular, the loss minimized is:

$$\mathcal{J}_{TP}(y; F_\theta(\mathbf{x}, t)) = \ell(y, F_\theta(\mathbf{x}, t)) + \lambda(\ell(y, F_\theta(\mathbf{x}, t + \delta))), \tag{5}$$

were $\delta$ is chosen adversarially through gradient ascent on $\ell(\cdot; \cdot)$

Table 2 summarizes the results. We make the following observations. (i) GI outperforms other forms of regularizations on most dataset. (ii) The poor performance as well as stability issues of Grad-Reg are due to the fact that a supervised gradient regularization makes the model more time agnostic. (iii) TimePerturb can also provide significant gains since the method is designed to provide robustness against small perturbations of inputs. GI also perturbs time, but the key difference is that TimePerturb assumes that $y$ stays the same after perturbation. In constrast, GI makes a milder linearity assumption on the computed logits allowing GI to explore larger perturbation windows without introducing undue smoothness bias.

## 4.6 Evaluation of time-sensitive architecture of our model

There are many ways to make the network parameters a function of time. We chose to modify only the ReLU units of the network and make their activation a function of time using our TReLU module. Here, we investigate the impact of these units through an ablation study. We start with a network with only ReLU activation functions. We then change the last few layers increasingly to be time-dependent and finally train a network using ERM to understand how well our temporal parameterization does on its own. Figure 3 summarizes the results. We observe that the performance becomes better with the number of time dependent units. Even a single TReLU unit gives significant gain against a time-insensitive architecture, indicating its effectiveness in capturing temporal trends.

## 4.7 Accuracy Along Time

We compare the efficiency of GI with CIDA and Adagraph by training the model using only limited number of time snapshots on the Rot-MNIST dataset. We do this by fixing the test time snapshot $T$, and training independent models on data from times $\{T - 1\}$, $\{T - 1, T - 2\}$ and so on. Figure 4 summarizes the results, which shows that GI works better especially in the low data regime, though the performance gap closes as the number of train time-stamps increases. It is also notable that performance across all methods is improved by including older training data, indicating the importance of harnessing longer term temporal trends in the data.

## 4.8 Impact of adversarial selection of $\delta$

In this section, we present a comparative analysis of various $\delta$ selection schemes. Adversarially selecting $\delta$ incurs additional overhead with regards to running time of our algorithm. We tried the following methods:

**Random** $\delta$: We randomly select a value of $\delta$ as $\delta \leftarrow \text{Unif.}[-\Delta, \Delta]$ for each mini-batch.
**Adversarial** $\delta$: Following Algorithm 1, we adversarially pick a $\delta$ for each mini-batch
**Adversarial** $\delta$ **- WS**: Same as above, but instead of sampling a $\delta$ for each mini-batch, we warm start, i.e. continue with the $\delta$ learnt in the previous training step.

From Table 3, it is clear that selecting $\delta$ adversarially according to Algorithm 1 provides some performance gains. However, warm-starting the value of $\delta$ from the previous training minibatch gives similar or better performance on most datasets. This reduces the overhead of sampling $\delta$ for each minibatch. Randomly selecting $\delta$ performs slightly worse, but can be used for situations in which the training process has time constraints.

| Dataset | 2-Moons | Rot-MNIST | M5-Hob |
|---|---|---|---|
| Random $\delta$ | $3.8 \pm 1.2$ | $8.0 \pm 1.0$ | $0.12 \pm 0.10$ |
| Adversarial $\delta$ | $3.6 \pm 1.5$ | $8.3 \pm 1.4$ | $0.08 \pm 0.05$ |
| Adversarial $\delta$ - WS | $3.5 \pm 1.4$ | $7.7 \pm 1.3$ | $0.09 \pm 0.03$ |

Table 3: Effect of adversarial $\delta$ selection on GI .

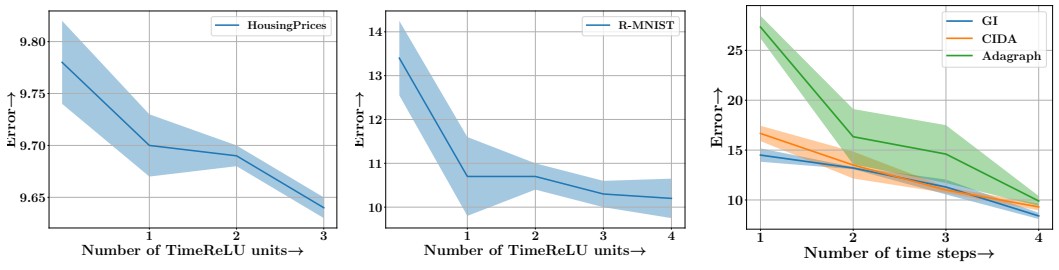

(a) House dataset     (b) Rot-MNIST dataset

Figure 3: Effect of changing the number of TReLU units

Figure 4: Effect of number of training snapshots.

## 4.9 Importance of number of finetune domains

When fine-tuning with GI, we finetune only on a subset of the training data, picking only $k$ most recent train time stamps. There is a trade-off inherent to this process, wherein one could pick a larger $k$ to have more training data, at the cost of higher training time. As observed in table 4, $k = 2$ performs comparably with higher values of k and at a lower computational expense, and hence we use this value for all our experiments.

| Dataset | Rot-MNIST | House | Elec2 |
|---|---|---|---|
| $k = 1$ | 8.6 | 9.9 | 17.2 |
| $k = 2$ | 7.7 | 9.6 | 16.9 |
| $k = 3$ | 8.0 | 9.7 | 16.9 |
| $k = 4$ | 7.9 | 9.6 | 17.3 |

Table 4: Effect of number of finetune domains on performance of GI .

## 5 Conclusions, Limitations and Future Work

In this work we identify the problem of generalizing prediction models for the future, and propose a simple method to address it. To that aim, we propose a Gradient Interpolation based approach which is agnostic of the deep network architecture used and can be used in various classification and regression tasks that exhibit data or label distribution shifts along time. Furthermore, we use TReLU and time representation vectors which can make any model robust to non-linear distribution shifts in time, like rotation. However, this is not without its shortcomings.

**Limitations.** One of the key elements in our proposed algorithm 1 is the running time overheads of searching for an adversarial $\delta$ in a $\Delta$ neighbourhood of the current timestamp. While we propose some engineering tricks to reduce this overhead in section A.4 of the appendix, it remains a significant component of the total training time of our algorithm. However, we note that the total wallclock training time of GI is still less than adversarial approaches like CIDA [5] due to faster convergence of our algorithm, as we detail in section B of the appendix.

**Future work** includes theoretically analyzing the GI loss, and extending GI loss to structured prediction tasks.

## 6 Acknowledgements

Vihari Piratla is supported by Google PhD fellowship. This research was partly sponsored by the IBM AI Horizon Networks - IIT Bombay initiative.

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
