# Supplementary Material
# Training for the Future: A Simple Gradient Interpolation Loss to Generalize Along Time

**Anshul Nasery** [*][†]     **Soumyadeep Thakur**[*]     **Vihari Piratla**     **Abir De**   **Sunita Sarawagi**
Department of Computer Science
Indian Institute of Technology, Bombay

In the main text, many algorithmic details were omitted and only discussed briefly. In this supplementary document, we expand the discussions in the main text and provide more details about the various datasets used, implementation details of the various methods, and some engineering tricks used. The source code for running our experiments can be found at this url[3]. We conclude this document with brief discussion of its broader impact.

## A  Experimental Details

### A.1  Dataset Details

We expand upon the seven datasets used for our experiments in this section.

**Rotated 2 Moons dataset:**  This is a variant of the 2-entangled moons dataset, with a lower moon and an upper moon labeled $0$ and $1$ respectively. Each moon consists of $100$ instances, and $10$ domains are obtained by sampling $200$ data points from the 2-Moons distribution, and rotating them counter-clockwise in units of $18°$. As a result a domain $i$ is rotated by $18i°$ degrees. Domains 0 to 8 (both inclusive) are our training domains, and domain 9 is for testing.

**Rotated MNIST dataset:**  This is an adaptation of the popular MNIST digit dataset, where the task is to classify a digit from $0$ to $9$ given an image of the digit. We generate $5$ domains by rotating the images in steps of 15 degrees. To generate the $i^{th}$ domain, we sample 1,000 images from the MNIST dataset and rotate them counter-clockwise by $15 \times i$ degrees. We take the first four domains as train domains and the fifth domain as test.

**Shuttle** [4] This dataset provides 9 features for about $58,000$ datapoints for space shuttles in flight. The task is multi-class classification with a heavy class imbalance. The dataset was divided into 8 domains based on the time-stamps associated with points, with times between 30-70 being the train domains and $70 - 80$ being the test domain.

**Electrical Demand (Elec2)**[5] This contains information about the demand of electricity in a particular province. It has 8 features including price, day of the week and units transferred. The task is, again binary classification, to predict if the demand of electricity in each period (of 30 mins) was higher or lower than the average demand over the last day. We discard instances with missing values. We consider two weeks to be one time domain, and train on 29 domains while testing on domain 30. There are hence $27,549$ train points and $673$ test points.

---

[*]Equal Contribution

[†]anshulnasery@gmail.com

[3]https://github.com/anshuln/Training-for-the-Future

[4]https://archive.ics.uci.edu/ml/datasets/Statlog+(Shuttle)

[5]http://web.archive.org/web/20191121102533/http://www.inescporto.pt/~jgama/ales/ales_5.html

35th Conference on Neural Information Processing Systems (NeurIPS 2021).

**Reutersdataset**[6]: This dataset contains text articles over a period from February 1987 to October 1987. The task involves predicting one of the 10 categories to which an article belongs. A total of 4297 articles from 26 Feb to 7 April are considered as training data, and they are split into 3 domains, each spanning roughly 14 days. 685 articles from 7 April to 23 April constitute the test data.

**House prices dataset**[7]: This dataset has housing price data from 2013-2019. It has 3 features: number of bedrooms, house type, postal code. This is a regression task to predict the price of a house given the features. We treat each year as a separate domain, but also give information about the exact date of purchase to the models. We take data from the year 2019 to be test data and prior data as training. There are $1,385$ test points and $20,937$ train points.

**M5: Household**[8]: The task is to predict item sales at stores in the state of California (US) given a history of sales of each product under the household category at each store. We use the monthly sales history from 2013 to 2015 for each item at each store. Our target domain is to predict sales of January 2016. We filter out those stores in the state of California, which have an average daily sales of less than 1 item. Each source domain has $124,100$ instances, and the target domain has $5,100$. We extract 78 features for each sales record, including rolling statistics over the past 15 days, information about holidays, day of the week, etc.

**M5: Hobbies** This is the same dataset as above but products are under the Hobbies category. The split between source and target domains, and the prediction task are also the same as the *M5: Household* dataset. Each source domain has $323,390$ instances and the target domain has $27,466$.

## A.2 Methods Compared

In this section, we provide further details of the baselines compared against in the main paper, as well as other methods we attempted.

### A.2.1 Details of Baselines

**CIDA** [5] : The authors take time as an input and aim to learn time-invariant feature representations using an encoder network. They make use of 3 deep neural networks - (1) Encoder: learns time-invariant features, (2) Predictor: predicts the class label given a time-invariant representation, (3) Discriminator: predicts the time-stamp given a time-invariant representation. We use the open-source code[9] provided by the authors, modifying it to suit regression tasks as needed.

**CDOT** [3]: In this method, the authors use regularized Optimal Transport (OT) maps to transform data from the source domain to the target domain. The authors propose adding a temporal regularization and a class regularization to the regularized OT training objective. The temporal regularization enforces a smoothness on the learnt Optimal Transport maps (couplings) along time. If we learn an OT coupling between 2 domains $\mathcal{D}^i$ and $\mathcal{D}^j$, the learnt coupling $I_{i \to j}(\mathbf{x})$ transforms $\mathbf{x} \in \mathcal{D}^i$ into its transported image in $\mathcal{D}^j$

The authors used this algorithm in a setting where only 1 labelled source domain is present. Since we have $T$ labelled source domains, we transform data from the last $k$ source domains to the target domain. For all datasets except *Elec2*, we set $k = T$. For the *Elec2* dataset, setting $k = 15$ gives the best results. Also, instead of using the class regularization, we force the OT coupling between 2 data points from different classes to be 0. We can do this because the true labels are known to us for all source data. We use the Python Optimal Transport library [10] to implement CDOT for our experiments.

**Adagraph** [9]: The authors use domain indices as inputs, using these to decide which batch-norm parameters to use for a particular input batch. We use an RBF kernel to compute similarity between different domains. We use their public source code[11] for our experiments.

---

[6]http://disi.unitn.it/moschitti/corpora.htm
[7]https://www.kaggle.com/htagholdings/property-sales
[8]https://www.kaggle.com/c/m5-forecasting-accuracy
[9]https://github.com/hehaodele/CIDA/
[10]https://pythonot.github.io/
[11]https://github.com/mancinimassimiliano/adagraph

### A.2.2 Other methods proposed

**DPerturb**: Inspired by methods such as [15], we have a baseline which attempts to augment the training data with perturbed examples. We train 2 separate models, a classifier $F(\mathbf{x}, t)$ which predicts the class label of the data point $\mathbf{x}$; and an Ordinal classifier $G(\mathbf{x}_1, \mathbf{x}_2)$ which predicts the temporal similarity between 2 data points $\mathbf{x}_1, \mathbf{x}_2$.

The classifier $F$ is first pre-trained on the entire source data. Parallely, the ordinal classifier $G$ is trained on pairs of source domains $\mathcal{D}^i, \mathcal{D}^j, i < j$. Using the gradients of this ordinal classifier with respect to the inputs, we perturb train examples to resemble examples from the test time domain. We finally fine-tune the pretrained classifier model using these simulated test examples.

**Deep ODE**: We train a deep neural ODE [**?** ] to model the evolution of an example data point along time. Neural ODEs parameterize the continuous dynamics of hidden units using an ordinary differential equation (ODE), which is specified by a deep neural network. We use this to transform examples from the train time into examples from the test time and augment the dataset with such examples. We use the public source code[12] by the authors for our experiments with this method.

It is important to note that Neural ODEs work best in a setting where we have images of the same data point across multiple domains. Hence, it works well only in toy settings like the *2-Moons* dataset, where the same sample is rotated along time. On real datasets however, we have different samples from each domain, and such a method cannot be applied directly. As a result we are able to compare Deep ODE to other models only for the *2-Moons* dataset.

**Generative Model**: We train a GAN-like generative model to transform an example datapoint from time $t_i$ into a datapoint from time $t_j$. To provide supervision to this model we use optimal transport maps to find the image of the data point at $t_j$. We use this to transform examples from the train time into examples from the test time and augment the dataset with such examples. The model consists of a "transformer" $G(\mathbf{x}, t_i, t_j)$, a discriminator $D(\mathbf{x}, t_j)$ and a classifier $C(\mathbf{x}, t_j)$. The transformer predicts the future image of a data point $x \in \mathcal{D}^i$ at $t_2$. The discriminator $D(\mathbf{x}, t_j)$ outputs 1 if $\mathbf{x}$ is a true data point from the $j^{th}$ time stamp, and 0 otherwise. We use a combination of various losses to train the transformer and the discriminator.

- Adversarial Loss - This is the standard GAN training objective which distinguishes between the actual points from time $t_2$ and the generated points.
- OT reconstruction Loss - We compute the optimal transport maps $I_{k \to (k+1)}(\cdot)$ to transport examples from time-stamp $k - 1$ to time-stamp $k$. These are pre-computed using training data from adjacent time-stamps. While training $G$, we add a loss term $\|I_{t \to (t+1)}(\mathbf{x}) - G(\mathbf{x}, t, t + 1)\|_2^2$ to guide the generation process.
- Classifier Loss - This loss tries to ensure that the label assigned to the generated sample by $C$ is the same as the original sample. Formally, the loss is $\ell(y, C(\mathbf{x}))$

The above models can be easily extended to a regression setting.

We minimize a combination of these losses for the generator and the discriminator. The discriminator is trained on the following loss:
$$L_D = -\mathbb{E}_{\mathbf{x} \in \mathcal{D}^i}[\log D(\mathbf{x}, i)] - \mathbb{E}_{\mathbf{x} \in G(\mathbf{z}, i-1, i)}[\log(1 - D(\mathbf{x}, i))]$$
The transformer is trained on:
$$L_G = -\mathbb{E}_{\mathbf{x} \in G(\mathbf{z}, i-1, i)}[\log D(\mathbf{x}, i)] + \lambda_1 \|I_{(i-1) \to i}(\mathbf{x}) - G(\mathbf{x}, i-1, i)\|_2^2 + \lambda_2 \ell(y, C(G(\mathbf{x}, i-1, i), i))$$

We first pre-train the classifier $C$ on all of the source data. Following that, we train the transformer and the discriminator in an adversarial manner. We fine-tune the classifier on the simulated target domain data points generated by the transformer.

Table 1 shows a comparison between these methods and *GI*.

We observe from Table 1 that *GI* performs well consistently. *DPerturb* works well for datasets where the data distribution changes noticeably. When the data distribution remains same, but the label

---

[12]https://github.com/rtqichen/torchdiffeq

| Dataset | 2-Moons | Rot-MNIST | M5-Hob |
|---|---|---|---|
| GI | $3.5 \pm 1.4$ | $7.7 \pm 1.3$ | $0.09 \pm 0.03$ |
| DPerturb | $5.6 \pm 2.6$ | $13.4 \pm 3.0$ | $0.28 \pm 0.09$ |
| Generative Model | $4.6 \pm 1.6$ | $13.5 \pm 3.2$ | $0.42 \pm 0.18$ |
| Deep ODE | $2.2 \pm 0.8$ | - | - |

Table 1: Comparison of GI against some other methods on some datasets in terms of misclasssication error (in %)for first 2 datasets and mean absolute error (MAE) for last dataset. The standard deviation over five runs follows the ± mark.

distribution changes across domains, the Ordinal Classifier fails to predict the difference in time-stamps between the data points. *Deep ODE* is quite powerful at capturing the continuous dynamics of the *2-Moons* dataset, however it cannot be applied to other datasets as discussed above. *Generative Model* can work somewhat well on synthetic data, however it fails to capture the transformations on real world data, making it inferior to GI .

### A.3 Network parameters and hyper-parameters

In this section we specify the architecture as well as other details for each dataset's experiments. We use Adam optimizer for all our experiments. In general we tune the learning rate individually for each dataset and method, picking from values between 1e-4 and 1e-2. We also tune $\lambda$ separately for GI, GradReg, TimePerturb picking from 0.01,0.1,0.5,1. We tune the hyper-parameters associated with selecting $\delta$ including $\Delta$ (between 0.1 and 0.5) and the number of steps (between 5-20) for GI, and use these same hyperparameters for all our ablations. These hyperparameters also ensure that the $\delta$ selection process converges for mini-batches. For AdaGraph we additionally tune the hyperparameters associated with the RBF kernel for domains.

**2-Moons**: For pre-training, we use a learning rate $lr = 5 \times 10^{-3}$, a fine-tuning $lr = 5 \times 10^{-4}$. We pretrain for 30 epochs and finetune for 25 epochs, early stopping during fine-tuning according the loss on the next domain. We finetune on the last two domains. For optimizing $\delta$ in the GI loss, we use vanilla gradient ascent with $lr = 5 \times 10^{-2}$ and $\Delta = 0.5$ The network architecture consists of 2 hidden layers, with a dimension of 50 each. We use a TReLU layer after each hidden layer, and use a Time2Vec representation with $m = 8$ and $m_p = 2$.

**Rot-MNIST** - For training the network, we use a pre-training learning rate $lr = 10^{-3}$, a fine-tuning $lr = 5 \times 10^{-4}$. We pre-train for 60 epochs and fine-tune for 20 epochs with early stopping. We fine-tune on the last two domains. For optimizing $\delta$ we use gradient ascent with $lr = 0.1$, $\Delta = 0.15$ and do 15 steps of this ascent. We use a ResNet like architecture with 4 CNN blocks having 16, 32, 64, 128 channels respectively, a kernel size of 3, followed by 2 fully connected layers of 256 and 10 units. We have a TReLU after each layer, and use Time2Vec with $m = 16$, $m_p = 4$.

**House** - For training the network, we use a pre-training learning rate $lr = 1 \times 10^{-3}$, a fine-tuning $lr = 5 \times 10^{-4}$. We pre-train for 40 epochs and fine-tune for 20 epochs with early stopping. We fine-tune on the last two domains. For optimizing $\delta$ we use gradient ascent with $lr = 0.3$, $\Delta = 0.2$ and do 5 steps of this ascent. We use a three layer neural network with a hidden size of 400. We have a TReLU after each layer, and use Time2Vec with $m = 16$, $m_p = 4$.

**Shuttle** - For training the network, we use a pre-training $lr = 5 \times 10^{-3}$, a fine-tuning $lr = 5 \times 10^{-4}$. We pre-train for 25 epochs and fine-tune for 15 epochs. We fine-tune on the last two domains. For optimizing $\delta$ we use gradient ascent with lr=$5 \times 10^{-3}$, $\Delta = 0.2$ and do 10 steps of this ascent. We use a two layer neural network with a hidden size of 128. We have a TReLU after each layer, and use Time2Vec with $m = 16$, $m_p = 4$.

**Reuters** - For training the network, we use a pre-training $lr = 2.0$, a fine-tuning $lr = 1.0$. We pre-train for 50 epochs and fine-tune for 20 epochs. We fine-tune on the last two domains. For optimizing $\delta$ we use gradient ascent with lr=0.1, $\Delta = 0.2$ and do 5 steps of this ascent. We use an Embedding bag layer, followed by a two layer neural network with a hidden size of 128. We have a TReLU after each layer, and use Time2Vec with $m = 8$, $m_p = 2$.

**Elec2** - For training the network, we use a pre-training $lr = 5 \times 10^{-3}$, a fine-tuning $lr = 5 \times 10^{-4}$. We pre-train for 30 epochs and fine-tune for 20 epochs. We fine-tune on the last two domains. For optimizing $\delta$ we use gradient ascent with lr=$5 \times 10^{-3}$, $\Delta = 0.2$ and do 10 steps of this ascent. We use a two layer neural network with a hidden size of 128. We have a TReLU after each layer, and use

Time2Vec with $m = 16$, $m_p = 4$.

**ONP** - For training the network, we use a pre-training $lr = 10^{-3}$, a fine-tuning $lr = 5 \times 10^{-5}$. We pre-train for 50 epochs and fine-tune for 30 epochs with early stopping. We fine-tune on the last 2 domains. For optimizing $\delta$ we use gradient ascent with lr=0.5, $\Delta = 0.1$ and do 10 steps of this ascent. We use a two layer neural network with a hidden size of 200. We have a TReLU after each layer, and use Time2Vec with $m = 8$, $m_p = 2$.

**M5-Hob** - For training the network, we use a pre-training lr=$10^{-2}$, a fine-tuning lr=$5 \times 10^{-4}$. We pre-train for 25 epochs and fine-tune for 15 epochs with early stopping. We fine-tune on the last 2 domains. For optimizing $\delta$ we use gradient ascent with lr=0.5, $\Delta = 0.5$ and do 5 steps of this ascent. We use a three layer neural network with a hidden size of 50. We have a TReLU after each layer, and use Time2Vec with $m = 8$, $m_p = 2$.

**M5-House** - For training the network, we use a pre-training $lr = 10^{-2}$, a fine-tuning $lr = 5 \times 10^{-4}$. We pre-train for 35 epochs and fine-tune for 20 epochs with early stopping. We fine-tune on the last 2 domains. For optimizing $\delta$ we use gradient ascent with lr=0.5, $\Delta = 0.5$ and do 5 steps of this ascent. We use a three layer neural network with a hidden size of 50. We have a TReLU after each layer, and use Time2Vec with $m = 8$, $m_p = 2$.

### A.4   Other Implementation Details for GI

**Finetuning details.** In the second step of our training Algorithm, i.e. finetuning with the GI loss, we finetune the model only with the last $k$ domains, where $k$ depends on the dataset. We do this to bias our network more towards *extrapolating* to future data rather than just interpolating between training domains.

Further, selecting $\delta$ adversarially is the step which takes up the most amount of time. Hence we use the following tricks to decrease this time-

- We use a single $\delta$ per minibatch, instead of having a separate $\delta$ for each example. We notice that this increases the performance, while providing a significant reduction in training time.
- We exit the $\delta$ training loop once the gradient falls below a threshold. We also initialize the value of $\delta$ for the next mini-batch using the value obtained from the previous mini-batch, to warm start the process.

Apart from this, we do early stopping during the fine-tuning step based on the actual loss $\ell$ incurred on the next domain. This is achieved by computing the value of $\ell$ on domain $j + 1$ while fine-tuning with $\mathcal{J}_{GI}$ on domain $j$.

**Details about neural architecture.** In order for our architecture to be more memory efficient, we introduce a variant of TReLU for CNNs, where the threshold and slope for all the units of a channel are tied together, reducing the number of outputs of the TReLU to be equal to the number of output channels.

### A.5   Hardware Setup

All experiments were run on dual Intel® Xeon® Silver 4216 Processors. All experiments on Rot-MNISTdataset were executed on a single Nvidia Titan RTX GPU.

## B   Computation Time

While our training steps are computationally more expensive than the baseline ERM training, we find that the end to end training time for our approach is less than that of adversarial methods like CIDA, due to faster convergence. We report the total training times in seconds in table 2

## C   Comparison with online learning

In order to compare our method with online learning approaches, we experiment with modified versions of FTPL and FTRL algorithms from [23] in our setting, by training the learner on time upto $t_T$ and testing it on time $t_{T+1}$. We modify the algorithm to output the weights of a neural network at

| Algorith | ERM | CIDA | GI |
|---|---|---|---|
| 2-Moons | 6.2 | 40.4 | 7.1 |
| Rot-MNIST | 45.5 | 257.3 | 207.1 |
| House | 91.4 | 352.4 | 195.6 |
| M5-Hob | 745.5 | 2892.6 | 1334.2 |

Table 2: Training time in seconds of various algorithms on some datasets.

time $t + 1$, and consider $f_i$ to be the cross entropy/MSE loss incurred by the model on time stamp $i$. We take $g_t$ to be $f_{t-1}$ and $\sigma_{t,j}$ to be the loss incurred on perturbed (gaussian noise added to input) points from domain $t - 1$. We set $m = 1$ as in the paper. We observe that GI performs better than both these baselines as reported in table 3

| Dataset | FTPL | FTRL | GI |
|---|---|---|---|
| Rot-MNIST | $11.5 \pm 3.1$ | $16.7 \pm 2.0$ | $\mathbf{7.7 \pm 1.3}$ |
| Elec2 | $18.5 \pm 1.7$ | $20.7 \pm 0.8$ | $\mathbf{16.9 \pm 0.7}$ |
| Shuttle | $0.75 \pm 0.12$ | $0.83 \pm 0.20$ | $\mathbf{0.21 \pm 0.05}$ |
| House | $10.2 \pm 0.1$ | $10.3 \pm 0.3$ | $\mathbf{9.6 \pm 0.02}$ |

Table 3: Comparison against online learning methods

## D   Potential Negative Societal Impact

Our algorithm can make models more robust to temporal shifts in data distribution. However, our method cannot handle large drifts. If used in human-facing applications such as credit rating we have to be aware of this limitation and exercise caution. Perhaps, the use of our method should be guarded with another module that detects the magnitude of drift. We note that this limitation is not unique to our approach. Any predictive model has the potential of making misleading predictions unless guarded with reliable out of distribution detection (OOD) and calibration modules.