# OpenReview forum: "Training for the Future: A Simple Gradient Interpolation Loss to Generalize Along Time"
_NeurIPS.cc/2021/Conference — NeurIPS 2021 Poster_

### Official Review · Reviewer_F2t2 · 2021-07-14

**Rating:** 6
**Confidence:** 4

**Summary:**

This paper presents the Gradient Interpolation (GI) loss to regularize a time-dependent network, aiming to generalizing the predictions to the near future. The GI loss is computed as the maximum of the loss within a short window, which encourages the network to be locally linear in time. The paper further designs the Time Leaky ReLU (TReLU) which uses the time feature to scale the standard ReLU functions. Through a list of classification and regression datasets, this paper demonstrates its superiority against baselines.

**Limitations And Societal Impact:**

To further improve the paper, I think these are potential directions,
1) A clearer discussion of Sec 3.2
2) Discussing the computational overheads by training deltas, and the parameter overheads by TReLU
3) A clear description on the workflow

 (a) how the pretraining is conducted

 (b) what is the training and the testing procedure

This paper does not lead to negative societal impacts.

**Main Review:**

### Overall Comment
To make the agent to extrapolate towards future, it is important to uncover the temporal dependencies. This paper attempts to learn a network that take in both the input and the timestamp. This paper presents the GI loss to regularizing the predictions of the network within a short time window. However, I am not fully convinced by the arguments and the analyses.

### The Synthetic Problem
Sec 3.2 is very difficult to understand for me .
1) I cannot understand the conditional distribution: $P(x_j | y, t) = P(x_j = y | t) = p_j (t) $. Why $x_j$ appears only in the first equation ?
2) Since a random guess is $50$%, the numbers $49$%, $55.4$%,$49.5$%  do not seems to statistically different to me. In consequence, I do not think this experiment makes a positive evidence. In fact, I don't think any simple neural network could uncover the one-to-one relationship between $ j $ and $ t  $ with data only in past domains.
4) I think a more proper gradient regularizer than Eq(2) is to regularizing the gradients over losses, e.g., $ \| \frac{ \partial l(y, x, t) }{\partial t } \|_2^2 $. This regularizer is also similar to the TimePerturb approach in Sec 4.5, which works comparably to GI.

### TRAIN DELTA
Given that the GI loss involves the gradient already, performing gradient descent to find $\delta$ requires evaluating the second-order derivatives. And multiple gradient descent updates are needed to find the optimum. Therefore, the resulting approach seems computationally very heavy. Could you also provide a running time comparison with the baselines in your experiments ?

### Pretraining
1) How the pretraining is done in an online setting, where tasks come one-by-one ?
1) Line 3-5 in Algorithm 1. Is the pretraining done one-by-one sequentially ? Given you assumed knowing all tasks, this does not seem necessary.

### Experiments
1) Could you present a clear definition of the training and the testing procedure ?
2) How many more trainable parameters are used in TReLU, compared to the standard network and the baseline approaches ?


### minor comments
1) Line 75: "we we "
2) Line 278: "the angle of rotations"
3) Line 301: "lower errors"
4) Line 305: "drifts"
5) Table 2: "performances", "the classification task", "the regression task"
6) Line 369: "Furthermore, we use ..... can make any model...."
7) Line 192: $ [0, 1]^d $, should use curly brackets for the binary case.
8) Line 196: $p_j(t), t\geq 3$
9) Line 262: I cannot understand the logic of this sentence: "we ideally require a universally approximator of time-dependent function", given that the proposed TReLU is not provenly universal.


**Time Spent Reviewing:**

4

---

> ### Author Response · Authors · 2021-08-10
> **Response to Reviewer F2t2**
>
> We thank the reviewer for their detailed and insightful comments. We address some of their concerns in the top level comment above, and address the rest below.
>
> The concerns regarding computational cost, train-test setup and number of parameters are addressed in the general comment. Please take a look.
>
> >I cannot understand the conditional distribution
>
> The conditional distribution in the synthetic setting section denotes that the binary variable $x_j$ takes the same value as the label $y$ with a probability $p_j(t)$, representing a time varying correlation between each feature and the label.
>
> > Line 3-5 in Algorithm 1. Is the pretraining done one-by-one sequentially ? Given you assumed knowing all tasks, this does not seem necessary.
>
> The pre-training steps (line 3-5 in Algorithm 1) are performed on the entire training data at once as explained in line 179 of the manuscript.
>
> > This regularizer is also similar to the TimePerturb approach in Sec 4.5, which works comparably to GI.
>
> Our regularizer in expr.(1) contains $\partial F/\partial t$ and can gain from regularizing the time variation of the logits (F(x, t)). We intended to show--with the alternate regularizer in expr (2)--that regularizing logit’s time gradient does not in itself explain the improvement.

---

> > ### Comment · Reviewer_F2t2 · 2021-08-25
> > **Response to Author Rebuttal**
> >
> > Thank the authors for the responses. After discussing with other reviewers, now I correct my preview comment that "any simple network cannot learn the toy problem". In fact, although the relationship between $ t $ and $ j -3 $ cannot be learnt, the network can still learn some knowledge about the task from the first two dimensions. Nevertheless, the toy problem is still worthy of a better presentation. For example, one might add a curve for the ground-truth approach. Alternatively, I believe that the authors can find another problem where the proposed approach has a clear advantage over the simple baselines, which could demonstrate the paper better.
> >
> > Furthermore, the paper can be clearer in terms of the problem formulation. Previously I thought that the problem is an online problem, where the agent learns and adapts over time. But now it seems that it is a standard supervised learning problem with time variables. I think it will be very beneficial to clarity this in a future revision.
> >
> > Finally, I will increase my score to 5.

---

> > > ### Author Response · Authors · 2021-08-31
> > > **New toy setup and dataset**
> > >
> > > We request the reviewer to look at the results of our experiments on another dataset posted as a comment here: https://openreview.net/forum?id=U7SBcmRf65&noteId=xrpvXyA2eVm
> > > We also present a second synthetic scenario that shows much bigger margins with the baseline as a general comment here: https://openreview.net/forum?id=U7SBcmRf65&noteId=lNQfqAfjiMc.
> > > If accepted, we will fix our presentation so that the problem statement is clearer.

---

> > > > ### Comment · Reviewer_F2t2 · 2021-08-31
> > > > **Responses**
> > > >
> > > > Thank the authors for the new experiments and new toy problems. I acknowledge the merits brought in the new results, where the proposed approach outperforms the baselines with a clearer evidence. I agree that the proposed approach can be a useful contribution for exploring the time dynamics in predictions. However, as also pointed out by other reviewers, the paper requires an improved writing and presentation, including the problem formulation, the toy demonstration and the model design choices. For now I think the pros overside the cons, and I would raise my score to 6.

---

### Official Review · Reviewer_rfub · 2021-07-16

**Rating:** 6
**Confidence:** 4

**Summary:**

ML models deployed in production pipelines are trained continuously to account for gradual shift in the data distribution. This paper deals with model generalization w.r.t. data in the near future in this continuous distribution shift. The paper proposes training a model with time-sensitive parameters where the temporal complexity is regularized by a Gradient Interpolation (GI) loss. Time-sensitive network architecture has two components: time-insensitive (basically standard neural network parameters) and time-sensitive parameters (which are the output of another small neural network which takes into account the time parameter). It also proposes a novel time dependent activation function, Leaky Temporal ReLU (TReLU), that allows for smooth changes along time dimension. Empirical evaluations demonstrate that the proposed scheme outperforms competitive baselines such as generative and adversarial approaches.

**Main Review:**


Strengths:
---------------

- Separation of the neural architecture into time-sensitive and time-insensitive parameters: (a) incorporating time as a feature using TIME2Vec embedding model, (b) time dependent feature processing using the proposed TReLU activation function.

- Gradient Interpolation loss to incorporate adversarial time dependence on the loss w.r.t. time-dependent component of the network.

Weakness:
---------------

- The underlying problem seems like an ideal setting where online learning should have been explored first.

- I have some reservations on the empirical evaluation of the scheme ( see below ). Barring the two synthetic classification datasets, the proposed scheme performs poorly than the baselines on the remaining classification datasets. Are the gains on the regression datasets mainly due to the fact that the time-representations of the input is better than the raw features?


Questions for Authors:
---------------
- Why can the underlying problem not be taken as an instance of an online learning? In online learning, at any time t, you play parameters W_t, and you receive the next data (which could even be adversarial in nature, specifically designed to fool W_t), and you update your parameters to adjust to this data? There are many learning schemes which have been deployed in this context such as Follow-The-Regularized-Leader, Follow-The-Perturbed-Leader, and other advanced variants of these schemes etc.

- What is the best one can hope for this continual setting? Since the assumption is that near future data may have no relation to the data recently seen. Could you expect any classifier to perform better than one jumbo network trained with all the data in the past (paper mentions that data in the past could be repeated in future)? Correct me if I'm missing some critical peace of this puzzle.

- The Toy example in the Sec.~3.2 with the synthetic setup is not convincing that the proposed scheme works as intended. Coin toss accuracy is 50%, ERM achieves 49%, GI (proposed scheme) achieves 55.4%, and scheme with gradient regularizer achieves 47.4%. Its not clear if this example can help you make any conclusion about any of these methods as they are very close to the random chance on this problem. Maybe this is not a good toy example to showcase the benefits of the proposed scheme.

- I'm a little lost in the empirical evaluation section. Given a dataset that has time components and a train/test split. How is the training done? How is the network evaluated? Is it just trained on the training set and simply evaluated on the test set? I expected the evaluation to be incremental where in you train with train data upto a certain time and then evaluate on the data past this time instance, and so on.

- Figure 4 is missing in the main text and I see no figure 4 in the appendix.

Writing Clarity:
---------------
- Suggestion to include an explicit assumptions paragraph, which states whether the near-by future has any correlation to what was seen recently or could it be related to anything in the distant past?

- Missing related works and connections with online learning

- Line 75: we we



**Time Spent Reviewing:**

7

---

> ### Author Response · Authors · 2021-08-10
> **Response to Reviewer rfub**
>
> We thank the reviewer for their detailed and insightful comments, and address their concerns below.
>
> > Why can the underlying problem not be taken as an instance of an online learning?
>
> The incremental fine-tuning baseline that we include in Table 1 is inspired by online learning, applied to the deep learning setting.  Pure online learning, (which in deep learning is also referred as continual learning) methods do not allow revisiting the training data, which is not true in our case.  Also, when data exhibits seasonality, online learning without the capability of revisiting data may suffer from catastrophic forgetting, which is a topic of central focus in the continual learning literature.
>
> > Could you expect any classifier to perform better than one jumbo network trained with all the data in the past (paper mentions that data in the past could be repeated in future)
>
> Training a time-insensitive network on all data in the past is clearly inferior as shown by the Baseline in Table 1. A time-sensitive jumbo network, on the other hand with a large number of parameters will be prone to overfitting on the past data itself, and might not provide good generalization on future time-stamps. The Base-Time model in Table 2 is precisely that.  We see that it is worse than GI.  We also note that the ERM algorithm in our synthetic setting achieves a training accuracy of 99%, while performing no better than random on the test time domain, while a model trained with GI can do better. We reiterate that our goal is to not to perform well on the past data, and rather on the future data
>
> >Figure 4 is missing in the main text and I see no figure 4 in the appendix.
>
> Figure 4 is beside Figure 3 (above line 341) in the main paper
>
> > Barring the two synthetic classification datasets, the proposed scheme performs poorly than the baselines on the remaining classification datasets. Are the gains on the regression datasets mainly due to the fact that the time-representations of the input is better than the raw features?
>
> ONP and Elec2 are two other classification datasets.  None of the methods are better than ERM on ONP, perhaps because the data does not have strong temporal trends.  On Elec2 we are significantly better than all other existing methods except CIDA. But CIDA is much worse than GI on five other datasets.
> Our gains are not just due to better time-representations.  If you compare GI with Base-Time which uses the same time-representation (in Table 2), GI provides gains on all five datasets. This goes to show that our gains are not just due to better time-representation, but also training with GI loss.

---

> > ### Comment · Reviewer_rfub · 2021-08-30
> > **Response to Author Rebuttal**
> >
> > Thank you for your feedback.
> >
> > I believe your response helps me better understand the paper. I do see the merit in the problem you are tackling as well as the importance of the two components proposed in this paper, namely TReLU activation and GI loss.
> >
> > Thank you for adding the result for another dataset, which instills some more confidence in the proposed method.
> >
> > I still have comments regarding the paper:
> > (a) The online learning scheme does not necessarily need to look at the just the immediate input instance, you are free to look at some meta history or look at the full history (which will be computationally costly but there exists method in the literature like FTRL and its advanced variants such as FTPL, etc.). This could help in tackling the time component of this scheme a little bit better.
> >
> > (b) Your toy example really needs a bit of thought. It introduces more questions than it answers. Specifically the part that it still stays near a random prediction. Its not clear how to judge any aspect of the proposed method or the tackled problem, except that you cannot simply rely on time component.
> >
> > (c) A better evaluation strategy would have been to simply look at the regret style metrics (wherein you learn the classifiers upto a timestep t and then evaluate on the data from t+1, and show the relative performance amongst all the baseline). This would keep your setup inline with the core problem (model drifting along time dimension).
> >
> > Having said this, I would be increasing my score to 6.

---

> > > ### Author Response · Authors · 2021-09-02
> > > **Response to comment.**
> > >
> > > We thank the reviewer for their valuable comments.
> > >
> > > a: Please note that our IncFineTune algorithm can be considered as an instance of FTL.  For non-convex learner’s like ours, we checked that recent deep learning methods (e.g. the Online Meta Learning paper of Finn et al[2]) call this as the FTL approach. We also experiment with modified versions of FTPL and FTRL (Alg. 2 of [1]) in our setting, by training the learner network on data of timestamps upto $t$ and testing it on time $t+1$. We modify the algorithm to output the weights of a neural network at time $t+1$, and consider $f_{i}$ to be the cross entropy/MSE loss incurred by the model on time stamp $i$. We take $g_t$ to be $f_{t-1}$ and $\sigma_{t,j}$ to be the loss incurred on perturbed (gaussian noise added to input) points from domain $t-1$. We consider $l_2$ regularization of the model weights for FTRL. We observe that GI performs better than both these baselines as reported below -
> > >
> > > | Dataset  | FTPL          | FTRL         |GI            |
> > > |----------|---------------|--------------|--------------|
> > > |Rot-MNIST     |$11.5 \pm 3.1$|$16.7 \pm 2.0 $|$\mathbf{7.7 \pm 1.3} $|
> > > |Elec2        |$18.5 \pm 1.7$|$20.7 \pm 0.8 $|$\mathbf{16.9 \pm 0.7} $|
> > > |Shuttle    |$0.75 \pm 0.12$|$0.83 \pm 0.20$|$\mathbf{0.21 \pm 0.05}$|
> > > |House     |$10.2 \pm 0.1$|$10.3 \pm 0.3$ | $\mathbf{9.6 \pm 0.02}$|
> > >
> > > [1]- Sai Suggala, A. and Netrapalli, P., “Follow the Perturbed Leader: Optimism and Fast Parallel Algorithms for Smooth Minimax Games” https://proceedings.neurips.cc/paper/2020/file/fd5ac6ce504b74460b93610f39e481f7-Paper.pdf
> > >
> > > [2] - Finn, C., Rajeswaran, A., Kakade, S., and Levine, S., “Online Meta-Learning”,
> > >
> > >
> > > b: We present a second setting   of the synthetic dataset in the [general comment](https://openreview.net/forum?id=U7SBcmRf65&noteId=lNQfqAfjiMc) that explains why GI works while displaying a much bigger gap with baseline.
> > >
> > > c: Please note that our evaluation is in the regret setting.  We train the classifier upto a certain time t, and evaluate at time t+1.This is in line with the formal definition of our problem as outlined in lines 32-48. The training set for each dataset contains examples from the beginning upto time T, and we test our model on examples from the T+1 timestamp. In section 4.7 and fig 4 of the paper, we also vary the number of training domains and compare performance on the fixed test domain.

---

> ### Author Response · Authors · 2021-08-27
> **Additional empirical results**
>
> To address concerns about the performance of our method on real-world data, we have also added additional empirical results as a comment [here](https://openreview.net/forum?id=U7SBcmRf65&noteId=xrpvXyA2eVm) - https://openreview.net/forum?id=U7SBcmRf65&noteId=xrpvXyA2eVm.

---

### Official Review · Reviewer_M61X · 2021-07-24

**Rating:** 7
**Confidence:** 4

**Summary:**

The authors propose a novel loss (the gradient interpolation loss), paired with various architectural innovations and a training algorithm, in order to train models that generalize gracefully under distribution shifts that happen over time. The main contributions are as follows:
1. The gradient interpolation (GI) loss, a regularization term that penalizes the curvature of the learned network along time.
2. The TReLU activation, a novel activation whose slope and threshold are learned to vary wrt. time.
3. A training algorithm that uses the GI loss to train networks that are robust against time-dependent distributions shifts.

The contributions of the experiments are as follows:
1. The proposed method seems to perform better than a number of selected baselines and state-of-the-art domain adaptation methods.
2. Further insight is derived from synthetic experiments that aim to understand the mechanisms by which the proposed method achieves robustness against domain shifts.

**Limitations And Societal Impact:**

The limitations, which are closely intertwined with the weaknesses, are outlined above.

**Main Review:**

I believe that this is a good paper that makes a solid and interesting contribution on an important problem. I'm ready and inclined to increase my score based on the author response to the main weakness I list below.

The main strengths include:
1. The proposed approach is interesting and, as far as I know, novel.
2. The GI loss can, in principle, play well with arbitrarily complex models.
3. The fact that time-dependence of network parameters can be achieved only through modifying the activation functions is noteworthy. (I'm not sure if this is a novelty)
4. The paper does relatively extensive ablation studies and runs other synthetic experiments to gain insight into how the proposed method works.

___ The main weakness ___:
1. It's not clear how the hyper-parameters were tuned, both for the competing benchmarks and in the ablation studies. For example, was the learning rate separately tuned, albeit coarsely, for all different tasks on all different models? It's also important to learn more about how the competing approaches for picking the delta term (i.e. adversarial, random, adversarial with warm start etc.) were tuned, as well as the lambda coefficient of the gradient-penalty loss (Equation 2). In my opinion, if it turns out that selecting delta randomly from a reasonable distribution works as well as the adversarial version, I think that'd be a positive for the proposed method, as it'd remove the costly optimization step to optimize delta each step.
2. It is specified in the appendix that the fine-tuning steps, which is where the GI loss is enforced, is applied only to the last k time stamps of the dataset. This is, I think, an important design choice and should be moved to the main paper. Additional clarification on the importance of choosing k properly is also needed (it seems like most experiments used k=2. How would be performance be if you used k=1, or k=3? ). (Please correct me if you've actually discussed this in the main paper and I've missed it somehow)

Other weaknesses include:
1. The proposed method is expensive (due to the online selection of the delta term in Algorithm 1), as acknowledged by the authors. The tasks that the proposed method is applied on are relatively small scale, making it difficult to see how much of a scalability issue the cost of enforcing the GI loss really incurs. A more detailed discussion (perhaps in the appendix) on how much of the compute is spent on different parts of the algorithm would improve the paper, though I don't think this is crucial.
2. (This might not be a weakness - I might have misunderstood something about TReLUs) You mentioned that each TReLU has neural networks "h" and "g" that have different parameters. If a single TReLU is a replacement for a single ReLU, then that'd mean we have to train thousands, maybe millions, of small neural networks that are inherently unparallelizable on GPUs. Doesn't this lead to the algorithm being very inefficient? (Again, I might have misunderstood this part and this might not be a weakness)
3. The delta term is optimized across minibatches and, in some configurations of the algorithm, warm started from the result found in the previous mini-batch. These break the iid. assumption. This should be acknowledged, as it could lead to unexpected effects.
4. Overselling phrases like "[the proposed method] makes optimal use of the temporal information", as in line 77, should be avoided, unless the notion of optimality is defined properly and justified theoretically or empirically.
5. Further details on the "TrainDelta" step in line 11 of Algorithm 1 would be useful to have in the paper.




Questions to the authors:
1. Also, do you backpropagate through the steps to find delta (I assume not, but this would be useful to clarify).
2. How important is it to pick the capital delta terms (i.e. the window in which the perturbations are picked)?
3. Did you inspect how the learned slope and threshold values look like?
4. Is the x axis of Figure 3 "number of TimeReLU units", or "number of timeReLU layers"?
5. Could you clarify on which task Figure 4 was obtained on?

Writing, typos etc.
1. The paper is well written, well organized and easy to follow.
2. It has a small number of typos:
- line 75: "we" is duplicated.
- line 155: repeated ".."
- line 180: nerual -> neural
3. The math expression under line 196 is a big confusing.

====Edit after author response====
I thank the authors for their response. Acknowledging some of the limitations that were pointed out during the review process, I believe that the methods and results are strong/relevant enough to be of interest to the NeurIPS community working on domain generalization with extrapolation to near future. This is why favour acceptance.

I encourage the authors to improve upon their expository toy example to demonstrate the utility of their approach better, and in general try to gain a more mechanistic understanding for why their approach seems to have good empirical performance and how this compares with the ablation methods, such as regularizing the gradient. I also urge the authors to improve on writing to make the problem definition as well as key design decisions clearer to the audience.

**Time Spent Reviewing:**

6

---

> ### Author Response · Authors · 2021-08-10
> **Response to Reviewer M61X**
>
> We thank the reviewer for their detailed and insightful comments. We address their concerns below-
>
> > It's not clear how the hyper-parameters were tuned, both for the competing benchmarks and in the ablation studies.
>
> We adopt the leave-one-out training strategy for tuning. The training split from the last timestamp ($T$) is considered the validation set for hyperparameter optimization. We then train on the whole training data using the best hyperparameters, before testing it on data from time $T+1$.
>
> > If it turns out that selecting delta randomly from a reasonable distribution works as well as the adversarial version, I think that'd be a positive for the proposed method, as it'd remove the costly optimization step to optimize delta each step.
>
> We agree it’d be a positive, however, we found that is not the case in Section 2 of the appendix. When $\delta$ is picked u.a.r from $[-\\Delta, \\Delta]$, it is not as effective as when picked adversarially as shown in Table 2 of Appendix.
>
> > Additional clarification on the importance of choosing k properly is also needed (it seems like most experiments used k=2. How would the performance be if you used k=1, or k=3? ).
>
> We report the performance with different values of k for three datasets in the table below, showing the MAE for House and error rate for Rot-MNIST and Elec2 datasets. We note that k=2 performs comparably with higher values of k and at a lower computational expense.
>
> |Dataset| k=1 | k=2 | k=3 | k=4 |
> |------|------|-----|-----|-----|
> |Rot-MNIST|8.6|7.7|8.0|7.9|
> |Housing|9.9|9.6|9.7|9.6|
> |Elec2|17.2|16.9|16.9|17.3|
>
>
>  > Also, do you backpropagate through the steps to find delta
>
> We do not backpropagate through the steps. Once delta is computed adversarially, we use it as a scalar to compute the GI loss.
>
>  > Is the x axis of Figure 3 "number of TimeReLU units", or "number of timeReLU layers"?
>
> It represents the number of TimeReLU layers
>
> > Could you clarify on which task Figure 4 was obtained?
>
> As detailed in sec 4.7, we have obtained these results on the Rotating MNIST dataset.

---

> > ### Comment · Reviewer_M61X · 2021-08-16
> > **Thank you for your feedback**
> >
> > Thank you for your feedback.
> >
> > Could you elaborate a bit more on the hyperparameter tuning process? For example, which hyperparameters did you tune separately for each experiment, and which you left constant throughout? Specific information on how you tuned the rivalling approaches (especially those in your ablation studies) would be very helpful. For example, it’s not clear how you picked the value for \Delta in your experiments in Appendix 2 to see if random sampling of \delta would work as well or not (maybe I missed it?). Without this information, it’s hard to be sure whether the inferior performance of the random \delta sampling is due to a suboptimal choice of hyperparameters.

---

> > > ### Author Response · Authors · 2021-08-18
> > > **Clarifications on hyperparameter tuning**
> > >
> > > We have mentioned the hyper-parameters used for each dataset for GI in section 1.3 in the appendix. We tune the learning rate individually for each dataset and method, picking from values between 1e-4 and 1e-2. We also tune $\lambda$ separately for GI, GradReg, TimePerturb picking from {0.01,0.1,0.5,1}.
> > >
> > > We tune the hyper-parameters associated with selecting $\delta$ including $\Delta$ (between 0.1 and 0.5) and the number of steps (between 5-20) for each dataset for GI, and use these same hyperparameters for all our ablations in sec 4.5 and appendix 2. These hyperparameters also ensure that the $\delta$ selection process converges for mini-batches.
> > >
> > > For AdaGraph we additionally tune the hyperparameters associated with the RBF kernel for domains.

---

### Author Response · Authors · 2021-08-27
**Additional empirical results**

In order to further verify the efficacy of GI on real-world classification tasks, we tested our method on one additional dataset previously used in the online learning literature, namely the Shuttle dataset[1]. We modify the task to our setting by dividing the training data on the basis of the time-stamp associated with the examples, considering points up to time-stamp 70 to be in our train domains, and time-stamps 70-80 to be our test domain. The task is a multi-class classification problem. We find that our method significantly outperforms the considered baselines and other methods, providing further empirical evidence in support of our method. We report the full results below, reporting the per cent classification error (lower is better).

| Method | Classification Error |
| -----------|-------------|
|Baseline | $0.77\\pm0.10$|
|IncFineTune | $0.83\\pm0.07$|
|Base-Time | $0.61 \\pm 0.14$|
|IncFinetune-Time | $0.52 \\pm 0.12$|
|TimePerturb | $0.67 \\pm 0.06$|
|Adagraph | $0.47 \\pm 0.04$|
|CIDA | Did not Converge|
|GI | *$\\mathbf{0.29\\pm0.05}$*|



[1] - https://archive.ics.uci.edu/ml/datasets/Statlog+(Shuttle)

---

> ### Author Response · Authors · 2021-08-31
> **Alternate Toy Setup**
>
> Consider the following slightly different synthetic setting that conveys the merits of GI better.
> Recall that the input has five features with time-dependent label correlation. We change the label correlation of the first two features such that $p_1$(t) = 0.6+0.1t and $p_2$(t) = 0.6.
> We made no other changes.
> In this setting, the average accuracy (over 5 runs) of GI is 70.9 while the average accuracy of any other method: ERM, CIDA, gradient regularization is close to the random baseline: 50%.
> This setting, we believe, should clear up the confusion. We will revise this section of our paper upon acceptance.
>
> In the earlier setting, the average label correlation of the first two features: 0.4, 0.6 respectively, are close to random correlation (slightly better than random label correlation may not even be realized with only 300 train examples). On the other hand, the average label correlation of the last 3 distracting features is around 0.67. The high utility of the distracting features makes it hard to recover a classifier that relies on the first two features--rendering only slightly better than the random classifier for the future.
> In the new setting, the average correlation of the first feature (0.7) is higher than in the old setting (0.4).

---

### Decision · Program_Chairs · 2021-09-27

**Decision:**

Accept (Poster)

**Comment:**

The reviewers feel that this paper introduces an interesting and potentially useful approach to learning in changing environments. They consider the approach sensible. Various reviewers originally had concerns both about the computational cost and about the experimental validation (especially details of hyperparameter optimization and choices of tasks and baselines). After some discussion, the reviewers feel like the authors have addressed their main concerns, and all of them recommend acceptance. I concur.